# Pseudane-VII Regulates LPS-Induced Neuroinflammation in Brain Microglia Cells through the Inhibition of iNOS Expression

**DOI:** 10.3390/molecules23123196

**Published:** 2018-12-04

**Authors:** Mi Eun Kim, Inae Jung, Ju Yong Na, Yujeong Lee, Jaewon Lee, Jong Suk Lee, Jun Sik Lee

**Affiliations:** 1Department of Life Science, Immunology Research Lab, BK21-plus Research Team for Bioactive Control Technology, College of Natural Sciences, Chosun University, Dong-gu, Gwangju 61452, Korea; kimme0303@naver.com (M.E.K.); inae673@naver.com (I.J.); skwndyd@naver.com (J.Y.N.); 2Department of Pharmacy, College of Pharmacy, Molecular Inflammation Research Center for Aging Intervention, Pusan National University, Busan 46241, Korea; yujeong0713@nanver.com (Y.L.); neuron@pusan.ac.kr (J.L.); 3Biocenter, GyeonggidoBusiness & Science Accelerator (GBSA), Suwon, Gyeonggi-do 16229, Korea

**Keywords:** *Pseudoalteromonas* sp. M2, pseudane-VII, anti-neuroinflammation activity, microglia

## Abstract

We previously isolated pseudane-VII from the secondary metabolites of *Pseudoalteromonas* sp. M2 in marine water, and demonstrated its anti-inflammatory efficacy on macrophages. However, the molecular mechanism by which pseudane-VII suppresses neuroinflammation has not yet been elucidated in brain microglia. Microglia is activated by immunological stimulation or brain injury. Activated microglia secrete proinflammatory mediators which damage neurons. Neuroinflammation appears to be associated with certain neurological diseases, including Parkinson’s disease and Alzheimer’s disease. Natural compounds that suppress microglial inflammatory responses could potentially be used to prevent neurodegenerative diseases or slow their progression. In the present study, we found that pseudane-VII suppresses neuroinflammation in lipopolysaccaride (LPS)-stimulated BV-2 microglial cells and brain. Pseudane-VII was shown to inhibit the LPS-stimulated NO, ROS production and the expression of iNOS and COX-2. To identify the signaling pathway targeted by pseudane-VII, we used western blot analysis to assess the LPS-induced phosphorylation state of p38, ERK1/2, JNK1/2, and nuclear factor-kappaB (NF-κB). We found that pseudane-VII attenuated LPS-induced phosphorylation of MAPK and NF-κB. Moreover, administration of pseudane-VII in mice significantly reduced LPS-induced iNOS expression and microglia activation in brain. Taken together, our findings suggest that pseudane-VII may represent a potential novel target for treatment for neurodegenerative diseases.

## 1. Introduction

Neuroinflammation is associated with neurodegenerative diseases, including amyotrophic lateral sclerosis, multiple sclerosis, Parkinson’s disease, and Alzheimer’s disease [1]. Various barriers protect the central nervous system (CNS), which includes the brain and spinal cord. The blood-brain barrier (BBB) protects the brain and maintains brain homeostasis. Inflammation or infection of the BBB leads to the extensive migration of leukocytes that change the organization of tight junction complexes within the BBB. Cells, including microglia and macrophages, migrate into the BBB and produce inflammatory mediators, such as cytokines, chemokines, and free radicals. In the CNS, such inflammatory responses can cause nerve cell death and exacerbate brain injury [2].

Microglia are a type of macrophage that reside in the CNS and play an important role in neuroinflammation [3]. Microglia presents a series of changes in morphology and function after activated by an acute insult to the CNS [4]. Upon activation by brain injury or inflammatory pathogens exposure, microglia is activated, and releases various pro-inflammatory mediators, including NO, IL-1β, IL-6, and TNF-α [5,6]. The production and accumulation of these inflammatory mediators could not only further regulate neuroinflammatory response, but also injure neurons [7,8]. The expression of these inflammatory mediators is regulated by mitogen-activated protein kinases (MAPKs) [9]. The MAPK pathway has been shown to induce the expression of cytokines, including IL-1β and TNF-α, which are involved in inflammatory responses of the CNS [10]. Taken together, microglia mediated neuroinflammation might be a key event in the induction of neurons damage. Therefore, suppression of microglia-induced neuroinflammation may alleviate the symptoms and progression of neurodegenerative diseases.

Many compounds acquired from natural resources are metabolites in the biosynthetic pathways of terrestrial plants and microorganisms. Plants, animals, and bacteria generate metabolites that belong to the quinolone class [11]. In previous studies, we used liquid chromatography-mass spectrometry to isolate 4-hydroxy-2-alkylquinoline (pseudane-VII) obtained from the wild-type marine bacterium *Pseudoalteromonas* sp. M2 [12]. Pseudane-VII is a secondary metabolite that has anti-melanogenic and inflammatory activity [12,13]. However, the anti-neuroinflammatory properties of pseudane-VII have not been examined.

In this study, we investigate whether pseudane-VII has anti-neuroinflammatory properties in vitro and in vivo, and whether this compound regulates the expression of inflammatory factors, including NO, iNOS, IL-1β, IL-6, and TNF-α.

## 2. Results

### 2.1. Pseudane-VII Inhibits LPS-Induced NO Production in BV-2 Microglial Cells

The MTT assay was used to measure the cytotoxicity of pseudane-VII in BV-2 microglial cells. We found that 0.5–5 μM pseudane-VII (Figure 1A) had no adverse effect on the cells (Figure 1B). To evaluate whether pseudane-VII regulates NO and ROS production, BV-2 microglial cells were pretreated with 0.5, 1, 2.5, or 5 μM pseudane-VII for 2 h and then stimulated with 200 ng/mL LPS for 22 h. The Griess reaction was then used to determine the level of NO produced. As shown in Figure 2, LPS stimulation resulted in a significant increase in NO and ROS production. Treatment of cells with pseudane-VII significantly reduced this LPS-induced NO and ROS production.

### 2.2. Pseudane-VII Decreases mRNA and Protein Levels of iNOS and COX-2

NO and ROS was free radicals that are synthesized by iNOS [14]. iNOS and COX-2 are major inflammatory factors. BV-2 microglia were incubated with 0.5, 1, 2.5, or 5 μM pseudane-VII, and LPS-induced expression of *iNOS* and *COX-2* mRNA was assessed. As shown in Figure 3A, pseudane-VII inhibited the expression of *iNOS* and *COX-2* mRNA in a dose-dependent manner. Using Western blot analysis, we found that pretreatment of cells with pseudane-VII significantly reduced iNOS and COX-2 protein levels in LPS-stimulated BV-2 microglia (Figure 3B). Pseudane-VII thus appears to inhibit the mRNA and protein expression levels of both iNOS and COX-2.

### 2.3. Pseudane-VII Inhibits mRNA Expression and Protein Production of the Pro-Inflammatory FactorIL-1β

Both mRNA and protein levels of the pro-inflammatory cytokines IL-1β, IL-6, and TNF-α were assessed. As shown in Figure 4A, pseudane-VII did not alter IL-1β, IL-6, and TNF-α mRNA expression in LPS-stimulated microglia. We next used ELISA to measure the protein production levels of several pro-inflammatory cytokines, including IL-1β, IL-6, and TNF-α. Interestingly, pseudane-VII significantly reduced IL-1β production levels but did not suppress IL-6 or TNF-α production levels (Figure 4B). These results indicate that the anti-neuroinflammatory effects of pseudane-VII are mediated via the regulation of IL-1β production.

### 2.4. Pseudane-VII Suppresses MAPK and NF-κB Phosphorylation

We used western blot analysis to determine whether pseudane-VII regulates the phosphorylation of MAPK and NF-κB in LPS-stimulated BV-2 microglia cells. As shown in Figure 5A, the phosphorylation of p38, ERK, and JNK increased between 5 and 30 min in LPS-stimulated BV-2 microglia. In cells treated with pseudane-VII, the LPS-stimulated phosphorylation of these proteins was suppressed. Next, we determined whether pseudane-VII inhibits NF-κB phosphorylation in LPS-stimulated BV-2 microglia. As shown in Figure 5B, pseudane-VII suppressed NF-κB-p65 phosphorylation at 60 min. To further pinpoint how pseudane-VII regulates such phosphorylation, we pretreated cells with inhibitors that target particular kinases, including SB203580 (which targets p38), SP600125 (which targets JNK1/2), and U0126 (which targets ERK1/2). BV-2 microglia was then stimulated with LPS in the presence or absence of pseudane-VII, and NO production was measured. Pretreatment of cells with the p38 inhibitor SB203580 and JNK1/2 inhibitor SP600125 resulted in strong inhibition of NO production compared to pretreatment of cells with the ERK1/2 inhibitor U0126. We found that ERK is a more important regulator than p38 MAPK and JNK1/2. These results indicate that pseudane-VII suppresses all MAPKs but primarily inhibits the ERK1/2 pathway (Figure 5C).

### 2.5. Pseudane-VII Ameliorates LPS-Induced iNOS Expression in Brain Tissue

To assess the anti-neuroinflammatory properties of pseudane-VII in vivo, mice were intraperitoneally (i.p.) injected with LPS followed by pseudane-VII pretreatment. Expression levels of iNOS and the microglia activation marker such as Iba-1were then measured. As shown in Figure 6A, pseudane-VII significantly suppressed iNOS expression in brain tissue. Increased Iba-1 signal and enlarged cell body were detected in CA1, CA3, and dentate gyrus (DG) region of LPS administered mice brain. However, pseudane-VII treatment significantly reduced Iba-1 fluorescence intensity on microglia compared to LPS treatment alone. These results suggest that pseudane-VII inhibited microglia activation in brain hippocampus (Figure 6B).

## 3. Discussion

Several lines of evidence confirm that neuroinflammation was closely involved in the pathogenesis of neurodegenerative disease, such as Alzheimer’s and Parkinson’s disease [15,16,17]. The brain resident immune cells, such as microglia, play a crucial role in the neuroinflammation. Therefore, many researchers have been studying the regulation of microglia activation or microglia-mediated inflammatory factors to control neuroinflammation.

In the present study, we demonstrate that pseudane-VII suppresses neuroinflammatory responses of LPS-stimulated BV-2 microglia both in vitro and in vivo. Pseudane-VII suppresses the LPS-induced production of pro-inflammatory mediators such as NO, ROS and IL-1β in BV-2 microglia. Additionally, pseudane-VII inhibits not only the LPS-induced phosphorylation of MAPK but also that of NF-κB. Furthermore, pseudane-VII attenuates LPS-induced iNOS expression and activation of microglia in the brain. Therefore, our findings show that pseudane-VII has strong anti-neuroinflammatory effects. Microglia plays an important role in mediating various neuroinflammatory responses. Activated microglia induce NO and ROS production. Nitrite free radicals produced by excess levels of NO metabolites can have a deleterious effect on neurons [18]. It has been reported that NO production is mediated by iNOS [19]. For this reason, the regulation of iNOS expression plays an important role in neuroinflammation. As shown in Figure 2 and Figure 3, we found that pseudane-VII significantly inhibits NO production and suppresses the mRNA and protein levels of iNOS and COX-2 in LPS-stimulated BV-2 microglial cells. These results support that our previously reported pseudane-VII inhibited the gene and protein expression levels of iNOS and COX-2, which were increased by LPS stimulation in macrophage [12].

Many studies have suggested that microglia-mediated proinflammatory mediators, such as IL-1β, IL-6, and TNF-α, were recognized to contribute to neurodegeneration [20]. The overproduction of pro-inflammatory cytokines such as IL-1β, IL-6, and TNF-α in the brain can result in tissue damage and nerve cell death. These pro-inflammatory cytokines are released by activated microglia. Therefore, reduction of neuroinflammatory mediator production might present a promising therapeutic potential for neurodegenerative disease. For this reason, we examined the effects of pseudane-VII on the expression of these pro-inflammatory cytokines. We found that pseudane-VII selectively suppresses IL-1β protein levels in LPS-stimulated microglia (Figure 4). Interestingly, pseudane-VII selectively inhibited IL-1β in macrophage [12]. This suggests that pseudane-VII has an anti-inflammatory effect through the regulation of IL-1β in macrophage and microglia.

Previous reports have shown that MAPKs, including p38, JNK, and ERK, play a major role in IL-1β production in LPS-stimulated microglia [21]. Several studies have shown that ERK1/2 phosphorylation is important for IL-1β production. MAPK activation is linked to NF-κB activation via the Toll-like receptor 4. MAPK activates downstream transcription factors, including NF-κB, in microglia [22,23,24]. NF-κB activation of microglial cells induces the production of pro-inflammatory cytokines, neurotoxic reactive oxygen species, and excitotoxins [25,26,27]. By contrast, suppression of NF-κB activation reduced the pro-inflammatory factors release [4]. Therefore, we assessed whether psendane-VII regulates signaling pathways associated with MAKPs phosphorylation and NF-κB activation. As shown in Figure 5, pseudane-VII suppresses the phosphorylation of p38, JNK, and ERK1/2. To identify which signaling pathway is primarily targeted by pseudane-VII, we measured the amount of NO produced in LPS-stimulated microglia in the presence or absence of pseudane-VII after treatment with selective inhibitors of certain signaling pathways. We found that the anti-neuroinflammatory effects of pseudane-VII primarily involve the ERK1/2 pathway. Our results show that pseudane-VII has anti-neuroinflammatory efficacy through the inhibition of ERK1/2 phosphorylation, however, according to reports of Kawakowsky et al., ERK phosphorylation is beneficial to Alzheimer’s disease because it can induces neuronal survival and cognition [28,29]. Moreover, studies by Jahrlin et al., show that pERK is important for memory formation because the formation of PPARγ on the pERK protein complex is essential for memory formation [30]. Therefore, at least in the present study, although inhibition of ERK showed neuroinflammatory effects, ERK activity is more important for neuroprotection and neuronal cognition.

Furthermore, in vivo observations of the observed effects in vitro, pseudane-VII not only inhibited LPS-induced protein expression of iNOS in the brain but also markedly reduced the expression of Iba-1, a microglia activation marker (Figure 6A,B). These results suggest that pseudane-VII inhibits neuroinflammation both in vitro and in vivo.

In conclusion, our results demonstrated that pseudane-VII not only suppressed the production of NO and ROS, but also inhibited the expression of iNOS and COX-2, which are involved in the induction of NO. Moreover, pseudane-VII significantly inhibited pro-inflammatory cytokine production such as IL-1β in an LPS-stimulated BV-2 microglial cells.

Inhibition of NO and ROS, and IL-1β production was involved in the inhibition of MAPKs phosphorylation and NF-κB activation in LPS-stimulated BV-2 microglial cells. Therefore, our results indicate that pseudane-VII has anti-neuroinflammatory properties and could be used as a therapeutic agent applicable to microglia-related neuroinflammation.

## 4. Material and Methods

### 4.1. Chemicals and Reagents

Pseudane-VII obtained from Gyeonggi Bio-Center (Suwon, Gyeonggi-do, Korea). Griess reagent, 3-(4,5-dimethyl-2-thiazolyl)-2,5-diphenyl-2*H*-tetrazolium bromide (MTT), and TRI Reagent were purchased from Sigma-Aldrich (St. Louis, MO, USA). TNF-α, IL-1β, and IL-6 ELISA detection kit were purchased from BioLegend (San Diego, CA, USA). Antibodies were purchased from Santa Cruz Biotechnology (Dallas, TX, USA) and Cell Signaling Technology (Danvers, MA, USA). Dulbecco’s Modified Eagle’s Medium (DMEM), fetal bovine serum (FBS), and penicillin/streptomycin were purchased from WELGENE, Inc. (Gyeongsan-si, Gyeongsangbuk-do, Korea)

### 4.2. Cell Culture and Treatment

BV-2 murine microglial cell line were obtained from ATCC (Manassa, VA, USA) and cultured at 37 °C with 5% CO_2_ in DMEM supplemented with 10% FBS, 200 IU/mL penicillin, 200 μg/mL streptomycin, 4 mM L-glutamine, and 1 mM sodium pyruvate (complete medium). Pseudane-VII was dissolved in dimethyl sulfoxide (DMSO) and ethanol (a ratio of 1 to 9) and then diluted to required concentration with DMEM. And equal amount of DMSO and ethanol as final concentration of pseudane-VII were used as control sample.

### 4.3. Cytotoxicity Assay

Cytotoxicity was measured by colorimetric MTT assay. BV-2 (1 × 10^4^ cells/well) in complete medium were seeded into a 96-well cell culture plate; various concentrations of pseudane-VII (0 to 5 μM) were added to the wells, and the plate was incubated in 37 °C for 24 h. After treatment, medium containing pseudane-VII was removed and MTT (0.5 mg/mL) solution was added to each well. After incubation in 37 °C for 4 h, MTT solution was removed and the formazan product was dissolved in solvent (1:1 = DMSO:ethanol) resulting in a colored solution. Absorbance of the formazan solution was measured at 570 nm using an Epoch microplate reader (BioTek Instrument, Inc., Winooski, VT, USA).

### 4.4. NO Assay

BV-2 (2 × 10^4^ cells/well) in complete medium were seeded into a 96-well culture plate. Cultured cells were pretreated with various concentrations of pseudane-VII (0 to 5 μM) for 2 h, and then incubated in the presence or absence of LPS for 22 h. After incubation, the culture medium was mixed with an equivalent volume of 1× Griess Reagent and incubated for 15 min at room temperature. Absorbance was measured at 540 nm using a microplate reader (BioTech, Winooski, VT, USA).

### 4.5. RNA Isolation and RT-PCR

BV-2 microglial cells (3 × 10^5^ cells/well) were seeded in 12-well culture plate. Cultured cells were pretreated with various concentrations of pseudane-VII (0 to 5 μM) for 2 h, and then cells were incubated for 6 h in the absence or presence of LPS. After incubation, pseudane-VII-treated cells were collected by centrifugation and total RNA was isolated from the cells using TRI Reagent according to manufacturer’s protocol. To synthesize cDNA, 0.5 μg of total RNA was primed with oligo dT and reverse transcribed using a mixture of M-MLV RTase, dNTP, and reaction buffer (PromegaCorp, Madison, WI, USA). To measure the mRNA level of inflammatory factors including *GAPHD* (Forward; 5′-TGCACCACCAACTGCTTAG-3′, Reverse; 5′-GGATGCAGGGATGATGTTC-3′), *iNOS* (Forward; 5′-CTTGCCCCTGGAAGTTTCTC-3′, Reverse; 5′-GCAAGTGAAATCCGAT GTGG-3′), *COX-2* (Forward; 5′-TGGGTGTGAAGGGAAATAAGG-3′, Reverse; 5′-CATCATATTT GAGCCTTGGGG-3′), *il-1β* (Forward; 5′-GTGTCTTTCCCGTGGACCTT-3′, Reverse; 5′-TCGTTG CTTGGTTCTCCTTG-3′), *il-6* (Forward; 5′-CCTTCCTACCCCAATTTCCA-3′, Reverse; 5′-CGCAC TAGGTTTGCCCACTA-3′), and *tnf*-α (Forward; 5′-GGCCTCTCTACCTTGTGCC-3′, Reverse; 5′-TAGGCGATTACAGTCACGGC-3′), we designed the primers for target genes (Bioneer, Daejeon, Korea). The cDNA was amplified using the Gene Atlas G02 gradient thermal cycler system (Astec, Fukuoka, Japan) e-Taq DNA polymerase kit (Solgent; Daejeon, Korea) and the primers. The PCR products were visualized by RedSafe™ (iNtRON Biotechnology Inc. Seongnam-si, Gyeonggi-do, Korea) and a NaBI Gel-doc system (NEO Science; Suwon, Gyeonggi-do, Korea).

### 4.6. Enzyme-Linked Immunosorbent Assay (ELISA)

BV-2 microglial cells (2 × 10^4^ cells/well) were seeded in 96-well culture plates. Cells were pretreated with various concentration of pseudane-VII for 2 h, and then incubated in the absence or presence of LPS (200 ng/mL) for 22 h. IL-1β, IL-6, and TNF-α released into the culture supernatants was measured using Mouse IL-1β, IL-6, and TNF-α ELISA MAX™ Deluxe Sets (BioLegend, San Diego, CA, USA), according to manufacturer’s protocol. Briefly, standards and samples were added in capture antibody coated 96-well plate at 4 °C, overnight. After washing, detection antibody was incubated for 2 h and avidin-horseradish peroxidase (HRP) was reacted for 30 min. The wells were filled with substrate solution for 15 min and then stop solution (2 N H_2_SO_4_) terminated the reaction. Absorbance was read at 450 nm in a microplate reader.

### 4.7. Western Blot Analysis

BV-2 microglial cells (2 × 10^6^ cells/dish) were cultured in 60 mm culture dish. The cells were starved with serum-free DMEM for 4 h and then pretreated with pseudane-VII for 2 h, prior to treatment with LPS (200 ng/mL) for reaction time. The cells were collected and washed twice with cold 1X PBS and lysed with RIPA buffer (1% Triton X-100, 150 mM sodium chloride, 0.5% sodium deoxylcholate, 0.1% sodium dodecyl sulphate, 50 mM Tris, PH 8.0) supplemented with protease inhibitor cocktail (Sigma-Aldrich, St. Louis, MO, USA) and phosphatase inhibitor cocktail (GenDEPOT, Katy, TX, USA). The collected cells were homogenized using sonicator (Sonics & Materials, Inc., Newtown, CT, USA) and then the lysate cleared by centrifuging at 14,000× *g* for 15 min at 4 °C. The concentration of lysates was measured using Pierce™BCA protein assay kit (Thermo Fisher Scientific Inc., Waltham, MA, USA). Equal amounts of protein were separated by 10% SDS-polyacrylamide gel electrophoresis (SDS-PAGE) and transferred to a polyvinylidenedifluoride (PVDF) membrane. The membranes were soaked in the blocking solution (5% skim milk) for 1 h at room temperature. After washing three times, the membranes were incubated with primary antibodies overnight at 4 °C. HRP-conjugated anti-mouse IgG and anti-rabbit IgG antibodies were used as secondary antibodies. The detectable bands were visualized by enhanced chemiluminescence (ECL) western blot detection kit (ELPIS-Biotech Inc., Daejeon, Korea) and exposed to X-ray film.

### 4.8. In Vivo Experiment

All experiments were approved and performed in accordance with the regulations of the Chosun University Care and Use Committee (IACUC 2018-S0032). All mice were bred and housed at the Chosun University, Specific pathogen free (SFP) Animal Care Unit. C57/BL6 male mice were randomly divided in to 3 groups (Control, LPS, LPS + pseudane-VII). Pseudane-VII was dissolved in sterile saline and pre-administered by intraperitoneal injection (i.p.) (1 mg/kg) for 3 days prior to LPS injection. Also, LPS group was injected intraperitoneally (1 mg/kg) for 24 h. The expression of iNOS in brain was detected by using western blot.

### 4.9. Immunohistochemistry

The mice were intracardially perfused with 10 mM phosphate-buffered saline (PBS) and subsequently 4% paraformaldehyde. The brain was extracted and post-fixed in 4% paraformaldehyde for 24 h at 4 °C. Fixed brain was transferred to 30% sucrose solution. These brain samples were embedded in OCT for frozen sections, and then coronally sectioned at 40 μm using a freezing microtome (MICROM, Walldorf, Germany). Brain sections were washed with Tris-buffered saline (TBS; pH 7.5) and blocked with blocking solution (TBS-TS; TBS containing 0.1% Triton X-100 and 3% goat serum) for 30 min at room temperature. Then the sections were incubated with anti-ionized calcium-binding adapter molecule 1 (Iba-1) antibody (Wako Chemical USA, Inc., Richmond, VA, USA) in TBS-TS at 4% overnight. Brain sections were washed with TBS and incubated with anti-mouse IgG labeled with Alexa Fluor 568 for 3 h at room temperature. Stained sections were mounted with aqueous/dry mounting medium (Biomeda Corp., Foster, CA, USA). Confocal fluorescence images were acquired FV10i fluoview confocal microscope (Olympus, Tokyo, Japan).

### 4.10. Intracellular ROS Analysis

Treated cells were rinsed twice with 1× PBS and treated with 10 μM DCFDA (Molecular Probes, Leiden, The Netherlands) for 30 min. Then, the cells were harvested and washed with 1× PBS once. At last the cells were resuspended in 1% paraformaldehyde/PBS and FL-1 fluorescence was measured by FC500 Flow cytometer (Beckman Coulter, Brea, CA, USA). ROS production was expressed as mean fluorescence intensity (MFI), which was calculated using the Flowjo software V10.0 (Ashland, OR, USA).

### 4.11. Statistical Analysis

The results are presented as the mean ± SE (standard deviation). The results were analyzed by one-way analysis of variance (ANOVA) followed by LSD’s *post-hoc* test. The differences were considered statistically significant at * *p* < 0.05 and ** *p* < 0.001. The analysis was conducted using SPSS Statistics 24.0 (IBM; Armonk, NY, USA).

## Figures and Tables

**Figure 1 molecules-23-03196-f001:**
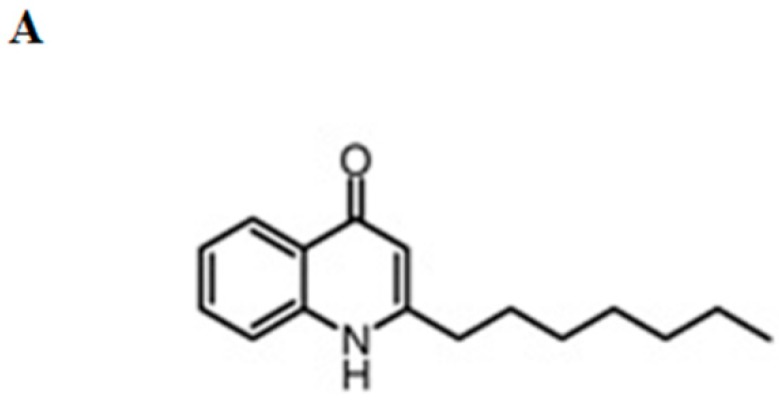
Structure of pseudane-VII and cytotoxic effect in BV-2 microglial cells. (**A**) Structure of pseudane-VII. (**B**) BV-2 microglial cells (1 × 10^4^ cells/well) were seeded at 96-well culture plates and then incubated at 37 °C in presence of 5% CO_2_. After incubation, cells were pretreated with various concentrations (0.5, 1, 2.5, and 5 μM) of pseudane-VII for 2 h and were added with LPS (200 ng/mL) for 22 h. The Control added with 0.025% solvent (DMSO:ethanol = 1:9) for 24 h. Cell viability was measured by MTT assay. The result is representative of repeated three independent experiments. Experimental results were indicated as mean (±SE).

**Figure 2 molecules-23-03196-f002:**
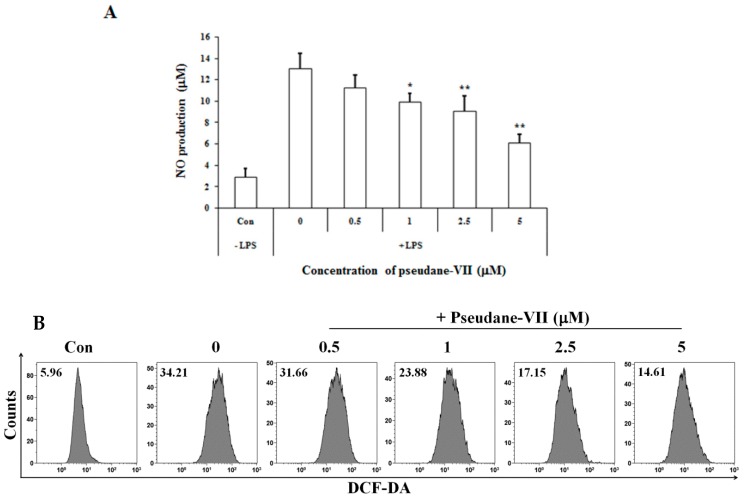
Pseudane-VII inhibits LPS-induced NO and ROS production in BV-2 microglial cells. Cells (2 × 10^4^ cells/well) were cultured in 96-well culture plates at 37 °C in presence of 5% CO_2_. Seeded cells were pretreated with various concentrations (0.5, 1, 2.5, and 5 μM) of pseudane-VII for 2 h and were added with LPS (200 ng/mL) for 22 h. (**A**) NO production was determined by Griess assay. Absorbance was measured at 540 nm using microplate reader. (**B**) ROS production was determined by FACS analysis using DCFDA. The result is representative of repeated three independent experiments. Experimental results were indicated as mean (±SE) (* *p* < 0.05, ** *p* < 0.001 vs. LPS-stimulated group).

**Figure 3 molecules-23-03196-f003:**
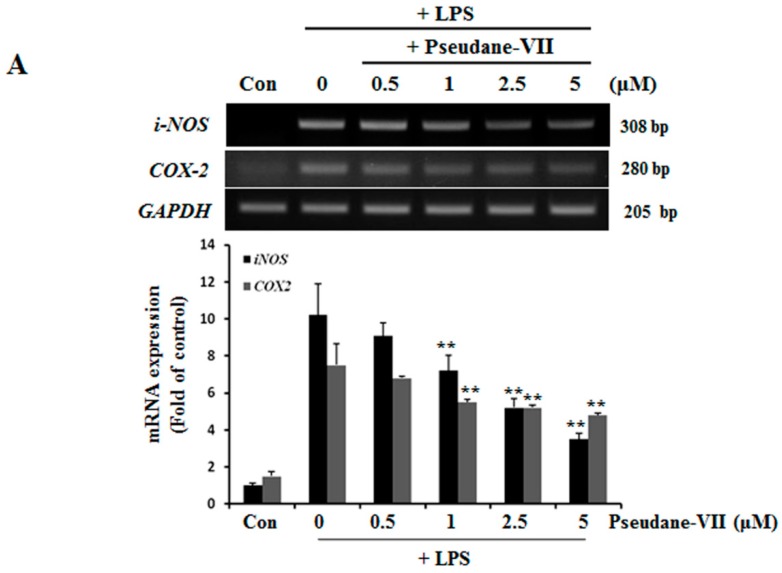
Pseudane-VII suppresses LPS-induced mRNA and protein level of iNOS and COX-2. (**A**) BV-2 microglial cells were treated with pseudane-VII (0.5, 1, 2.5, and 5 μM) for 2 h and then added with LPS (200 ng/mL) for 6 h. Expression levels of mRNA were confirmed by RT-PCR. GAPDH was used as internal control. (**B**) BV-2 microglial cells were pretreated with various concentrations (0.5, 1, 2.5, and 5 μM) for 2 h. After 2 h, cells were stimulated with LPS (200 ng/mL) for 24 h. An equal amount of cellular protein was separated by SDS-PAGE and transferred to the PVDF membranes. Membranes were attached to desired antibodies. β-actin was assessed as control. The result is representative of repeated four independent experiments. Experimental results were indicated as mean (±SE) (** *p* < 0.001 vs. LPS-stimulated group).

**Figure 4 molecules-23-03196-f004:**
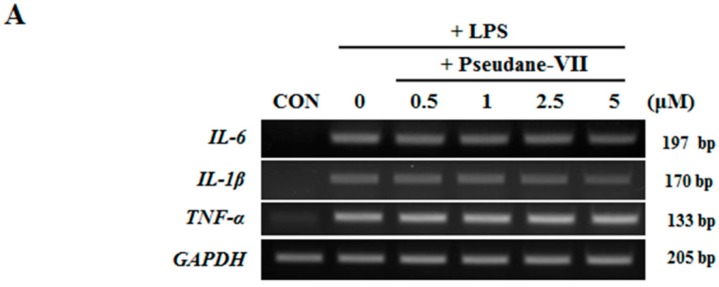
Pseudane-VII decreases LPS-induced pro-inflammatory cytokines. (**A**) BV-2 microglial cells were pretreated with diverse concentration (0.5, 1, 2.5, and 5 μM) of pseudane-VII for 2 h and then stimulated with LPS (200 ng/mL) for 6 h. Expression levels of mRNA were measured by RT-PCR. (**B**) Cells (2 × 10^4^ cells/well) were seeded in 96-well culture plates and then were pretreated with pseudane-VII (0.5, 1, 2.5, and 5 μM) for 2 h. Pretreated BV-2 microglia cells were added with LPS (200 ng/mL) for 22 h. Supernatants were collected for cytokine release. IL-1β, TNF-α, and IL-6 were measured by ELISA kit. Results are expressed as means (±SE) from three independent experiments. (** *p* < 0.001 vs. LPS-stimulated group).

**Figure 5 molecules-23-03196-f005:**
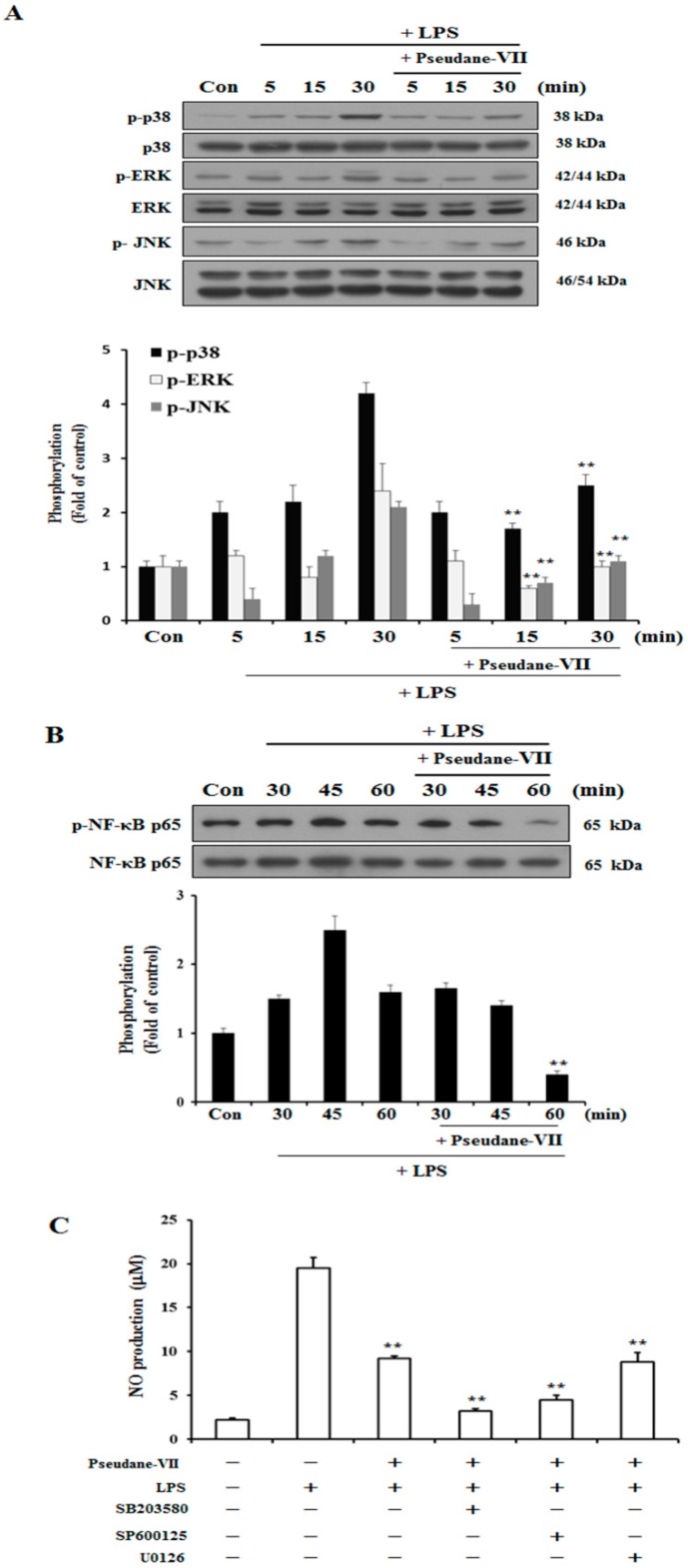
Pseudane-VII regulates LPS-induced phosphorylation of p38, ERK, JNK and NF-κB. (**A**,**B**) Cells were starved by using serum-free DMEM for 4 h. Next, cells were pretreated with 5 μM pseudane-VII for 2 h and stimulated with LPS (200 ng/mL) for 5, 15 and 30 min. Cells were lysed using RIPA buffer. Whole cell lysates was separated by SDS-PAGE and transferred to PVDF membranes. These membranes were attached to several antibodies (anti-p-p38, p38, p-ERK, ERK, p-JNK, and JNK antibodies). (**C**) The cells were pretreated with 10 μM of SB203580, SP600125, U0126 for 1 h and then added with 5 μM pseudane-VII for 2 h. After 2 h, the cells were stimulated with LPS (200 ng/mL) for 21 h. NO production was measured. Result was from representative experiments that confirmed identical patterns. (** *p* < 0.001 vs. LPS-stimulated group).

**Figure 6 molecules-23-03196-f006:**
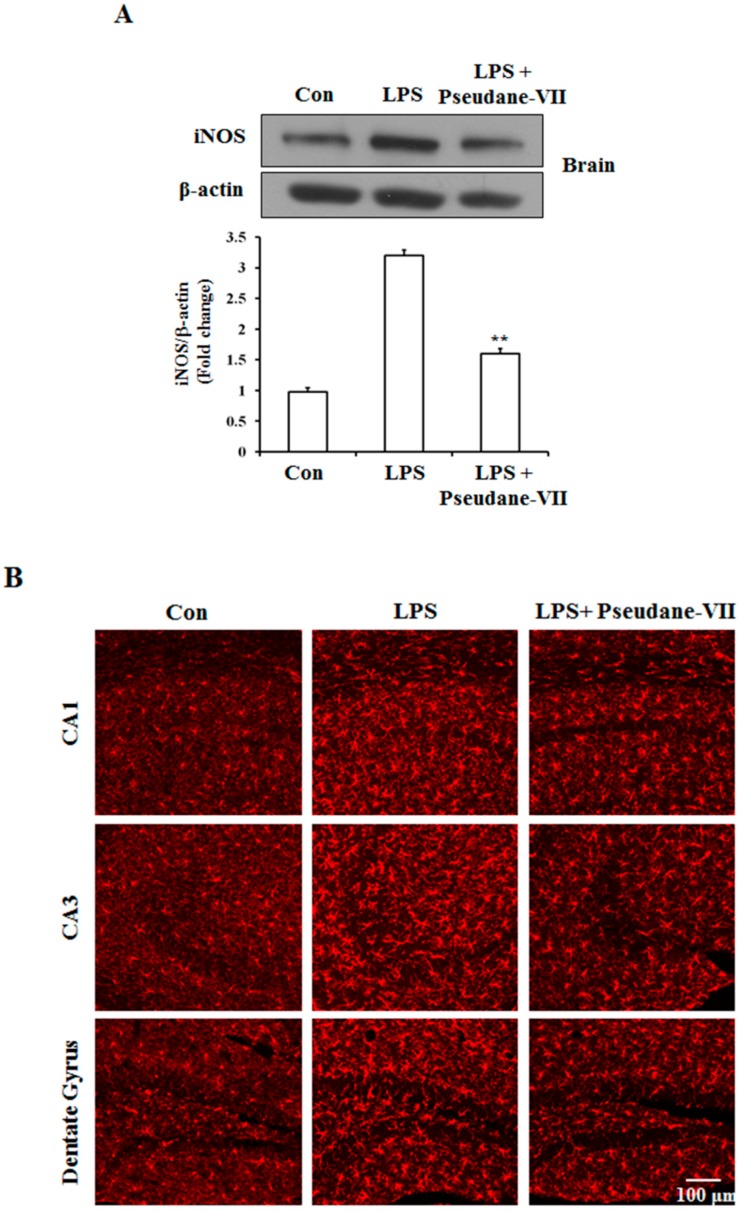
Pseudane-VII reduces the LPS-induced iNOS expression and microglial activation in brain hippocampus. C57BL/6 mice were randomly divided into three groups. The control group was challenged the same amount of solvent i.p. (*n* = 5). The treatment group was administered LPS (1 mg/kg) and pseudane-VII (1 mg/kg) as followed in vivo study design. 24 h after the injection, brain was harvested. Brain hippocampus was prepared protein lysates were used for iNOS detection (**A**). (**B**) Brain sections (40 μm) were immunostained with anti-Iba-1 (a microglia marker) antibody as described in the Materials and Methods. Iba-1 was increased in LPS administered mice brain compared to control. Pseudane-VII suppressed LPS-induced Iba-1-positive cells in CA1, CA3 and DG region of the hippocampus (Scale bar = 100 μm). Results are expressed as means (±SE) from three independent experiments. (** *p* < 0.01 vs. control group, ## *p* < 0.01 vs. LPS-stimulated group).

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
