# Peer review of "Pseudane-VII Regulates LPS-Induced Neuroinflammation in Brain Microglia Cells through the Inhibition of iNOS Expression"

_molecules, 2018, doi:10.3390/molecules23123196_

Round 1
Reviewer 1 Report
Dear Authors,
Please find below my personal recommendation about the manuscript.
General comments
The article concerns the anti-neuroinflammatory effects of 4-hydroxy-2-alkylquinoline (pseudane-VII) obtained from the wild-type marine bacteria Pseudoalteromonas sp. M2.
In the article (in lines 58-60), Authors declare that “here liquid chromatography-mass spectrometry was used to isolate” compound tested, however, any data were not shown in both methods' and results' sections. In contrary, in the chapter titled Chemicals and reagents, the information included is as follows: “Pseudane-VII obtained from Gyeonggi Bio-Center (Suwon, Gyeonggi-do, Korea) (line 239). Above mentioned discrepancy should be clarified. In my opinion, HPLC-MS analysis of pseudane-VII preparation and its purification should be characterized in the manuscript. Alternatively, the Authors should provide a reference to the article in which that data is included.
Anti-inflammatory effects of pseudane-VII were analyzed in both in vitro and in vivo neuroinflammation models with the use of lipopolysaccharide (LPS)-activated microglial BV-2 cells and C57/BL6 male mice administered with LPS by intraperitoneal injection.
In the first stage, pseudane-VII cytotoxicity was determined to non-activated BV-2 microglial cells by MTT test. No cytotoxic effects of pseudane-VII were observed at the dose range from 0.5 to 5 microM. Based on the obtained data, a dosage of pseudane-VII was established for the in vitro anti-inflammatory experiment. However, in the anti-inflammatory experiment, BV-2 microglial cells were exposed to pseudane-VII following LPS-stimulation. Thus, in the cytotoxicity studies, LPS-activated BV-2 cells should be treated with pseudane-VII not only the cells not activated by LPS. In my opinion, the cytotoxicity analysis was not carried out correctly according to the experimental conditions concerning the anti-inflammatory effect of the preparation. Therefore, the conclusion about the mitigation of the inflammatory response in microglial BV-2 cells without any cytotoxic effects induced appears to be non-confirmed by experimental results.
Whereas, data on the preparation effect on the proinflammatory enzymes (iNOS, COX-2), mediators (NO, intracellular ROS), as well as cytokines (TNF-a, IL-6 and Il-1b) analyzed by Real Time PCR, Western Blotting or ELISA, are quite well documented and described in the manuscript. The mechanism by which pseudane-VII down-regulates IL-1b mRNA expression and protein production is also well explained. In contrast, the inhibitory effects of pseudane-VII on the TNF-a and IL-6 expression were not observed in the LPS-activated BV-2 cells. I was wondering if the TNF-a mRNA analysis after cell activation was not too late. The TNF-a mRNA expression is usually analyzed by the first 2 hours following cell treatment with LPS and anti-inflammatory compounds. I would suggest, in the future, performing the screening analysis of the TNF-a mRNA expression in the cells stimulated by LPS and treated with pseudane-VII every 30 minutes for two hours.
The weakness of the experimental model used in this work is the lack of positive control. In my opinion, application of a reference substance with proven anti-inflammatory activity would allow better assess the anti-inflammatory potential of the test compound.
Specific comments
Introduction
Lines 58-60 – Authors declare: “Here, we used liquid chromatography-mass spectrometry to isolate 4-hydroxy-2-alkylquinoline (pseudane-VII) obtained from the wild-type marine bacterium Pseudoalteromonas sp. M2.” The manuscript does not contain any results of LC-MS analysis, nor there is any description in the method section. In the chapter 4.1. Chemicals and reagents, Authors declare that pseudane-VII was obtained from Gyeonggi Bio-Center (Suwon, Gyeonggi-do, Korea) (line 239). What's the truth?
Please, verify this information.
Results
2.1. Pseudane-VII cytotoxicity should be determined in the experimental inflammatory model with including BV-2 cell activation with LPS. It is significant for the assessment of NO production.
Figure 4B. The compound name (pseudane-VII ) in the X-axis descriptions should be corrected.
The figure caption, as follows “Pseudane-VII decreases LPS-induced pro-inflammatory genes and inflammatory cytokines” does not correlate to data presented. Pseudane-VII down-regulates only IL-1b cytokine expression in LPS-induced BV-2 cells, while TNF-a and IL-6 expressions are not affected by the compound tested. Please change the caption to a more appropriate.
2.5. I have a question about other pro-inflammatory mediators and cytokines in the in vivo experimental model. Were they determined in the study? Why are the results not shown in the manuscript? Please, explain this issue.
4. Material and Methods
4.2. Cell culture and treatment
Line 247 - Please provide the official source from which the cell lines were obtained (ATCC or ECACC or other cell culture collection). Give the catalogue number and the appropriate reference that is normally required.
4.3. Cytotoxicity assay
Line 256 – RAW264.7 macrophages were not used in this work. Please verify this information.
Intracellular ROS method description was not included in the Material and Method section. Please complete the missing methodology.
4.8. In vivo experiment
Line 321 - The i-NOS expression in the spleen were not presented in the manuscript.
Please, carefully check the spelling and editorial errors.
Author Response
RESPONSE TO REVIEWERS
Reviewer #1
General comments
The article concerns the anti-neuroinflammatory effects of 4-hydroxy-2-alkylquinoline (pseudane-VII) obtained from the wild-type marine bacteria Pseudoalteromonas sp. M2.
Comment 1: In the article (in lines 58-60), Authors declare that “here liquid chromatography-mass spectrometry was used to isolate” compound tested, however, any data were not shown in both methods' and results' sections. In contrary, in the chapter titled Chemicals and reagents, the information included is as follows: “Pseudane-VII obtained from Gyeonggi Bio-Center (Suwon, Gyeonggi-do, Korea) (line 239). Above mentioned discrepancy should be clarified. In my opinion, HPLC-MS analysis of pseudane-VII preparation and its purification should be characterized in the manuscript. Alternatively, the Authors should provide a reference to the article in which that data is included.
Response: We thank the reviewer for this helpful comment. We reported on the pseudane-VII structure in the previous paper. We attached a reference accordingly in lines 64-66.
Comment 2: Anti-inflammatory effects of pseudane-VII were analyzed in both in vitro and in vivo neuroinflammation models with the use of lipopolysaccharide (LPS)-activated microglial BV-2 cells and C57/BL6 male mice administered with LPS by intraperitoneal injection.
In the first stage, pseudane-VII cytotoxicity was determined to non-activated BV-2 microglial cells by MTT test. No cytotoxic effects of pseudane-VII were observed at the dose range from 0.5 to 5 microM. Based on the obtained data, a dosage of pseudane-VII was established for the in vitro anti-inflammatory experiment. However, in the anti-inflammatory experiment, BV-2 microglial cells were exposed to pseudane-VII following LPS-stimulation. Thus, in the cytotoxicity studies, LPS-activated BV-2 cells should be treated with pseudane-VII not only the cells not activated by LPS. In my opinion, the cytotoxicity analysis was not carried out correctly according to the experimental conditions concerning the anti-inflammatory effect of the preparation. Therefore, the conclusion about the mitigation of the inflammatory response in microglial BV-2 cells without any cytotoxic effects induced appears to be non-confirmed by experimental results.
Response: We thank the reviewer for this helpful comment. We determined that cytotoxicity was not observed by treating pseudane-VII and LPS together. Please see the attachment.
Comment 3: Whereas, data on the preparation effect on the proinflammatory enzymes (iNOS, COX-2), mediators (NO, intracellular ROS), as well as cytokines (TNF-a, IL-6 and Il-1b) analyzed by Real Time PCR, Western Blotting or ELISA, are quite well documented and described in the manuscript. The mechanism by which pseudane-VII down-regulates IL-1b mRNA expression and protein production is also well explained. In contrast, the inhibitory effects of pseudane-VII on the TNF-a and IL-6 expression were not observed in the LPS-activated BV-2 cells. I was wondering if the TNF-a mRNA analysis after cell activation was not too late. The TNF-a mRNA expression is usually analyzed by the first 2 hours following cell treatment with LPS and anti-inflammatory compounds. I would suggest, in the future, performing the screening analysis of the TNF-a mRNA expression in the cells stimulated by LPS and treated with pseudane-VII every 30 minutes for two hours.
The weakness of the experimental model used in this work is the lack of positive control. In my opinion, application of a reference substance with proven anti-inflammatory activity would allow better assess the anti-inflammatory potential of the test compound.
Response: We thank the reviewer for this helpful comment. We investigated gene expression at 6 hr, which is significantly increased by LPS in BV-2 microglia cells, but, in the future, we will perform at various times as recommended by reviewer. Although there are many compounds with anti-inflammatory activity, they are not used because they are difficult to use precisely as a positive control for anti-neuroinflammation.
Specific comments
Introduction
Comment 4: Lines 58-60 – Authors declare: “Here, we used liquid chromatography-mass spectrometry to isolate 4-hydroxy-2-alkylquinoline (pseudane-VII) obtained from the wild-type marine bacterium Pseudoalteromonas sp. M2.” The manuscript does not contain any results of LC-MS analysis, nor there is any description in the method section. In the chapter 4.1. Chemicals and reagents, Authors declare that pseudane-VII was obtained from Gyeonggi Bio-Center (Suwon, Gyeonggi-do, Korea) (line 239). What's the truth?
Please, verify this information.
Response: We thank the reviewer for this helpful comment. We added a reference on the structure of pseudane-VII in the Introduction section, and pseudane-VII was separated and analyzed with the collaborate with Gyeonggi Bio-Center.
Results
Comment 4: 2.1. Pseudane-VII cytotoxicity should be determined in the experimental inflammatory model with including BV-2 cell activation with LPS. It is significant for the assessment of NO production.
Response: We thank the reviewer for this helpful comment. We added a new data cytotoxicity of Psuedane-VII with LPS treatment in Figure 1B.
Comment 5: Figure 4B. The compound name (pseudane-VII) in the X-axis descriptions should be corrected.
Response: We thank the reviewer for this helpful comment. We changed the X-axis descriptions and added it in Figure 4B.
Comment 6: The figure caption, as follows “Pseudane-VII decreases LPS-induced pro-inflammatory genes and inflammatory cytokines” does not correlate to data presented. Pseudane-VII down-regulates only IL-1b cytokine expression in LPS-induced BV-2 cells, while TNF-a and IL-6 expressions are not affected by the compound tested. Please change the caption to a more appropriate.
Response: We thank the reviewer for this helpful comment. We changed mention of this in Result section.
Comment 6: 2.5. I have a question about other pro-inflammatory mediators and cytokines in the in vivo experimental model. Were they determined in the study? Why are the results not shown in the manuscript? Please, explain this issue.
Response: We thank the reviewer for this helpful comment. To assess the anti-neuroinflammatory properties, iNOS and microglia activation maker were analyzed from isolated hippocampus in brain. As a result of the isolation of hippocampus, it was difficult to analyze the production of pro-inflammatory cytokines by ELISA. Therefore, we identified iNOS, a representative inflammatory factor, and microglia activation as Iba expression.
4. Material and Methods
Comment 7: 4.2. Cell culture and treatment
Line 247 - Please provide the official source from which the cell lines were obtained (ATCC or ECACC or other cell culture collection). Give the catalogue number and the appropriate reference that is normally required.
Response: We thank the reviewer for this helpful comment. We added mention of this in 4.2 Cell culture and treatment of Materials and Methods section.
Comment 8: 4.3. Cytotoxicity assay
Line 256 – RAW264.7 macrophages were not used in this work. Please verify this information.
Response: We thank the reviewer for this helpful comment. We fixed it.
Comment 9: Intracellular ROS method description was not included in the Material and Method section. Please complete the missing methodology.
Response: We thank the reviewer for this helpful comment. We added intracellular ROS method description in Materials and Methods section.
Comment 10: 4.8. In vivo experiment
Line 321 - The i-NOS expression in the spleen were not presented in the manuscript.
Response: We thank the reviewer for this helpful comment. We fixed it.
Comment 11: Please, carefully check the spelling and editorial errors.
Response: We thank the reviewer for this helpful comment. We read carefully and fixed it.

Reviewer 2 Report
This manuscript describes the anti-neuroinflammatory effect of pseudane-VII isolated from Pseudoalteromonas sp. M2 on LPS-induced microglia in vitro and in vivo. Pseudane-VII inhibited LPS-induced phosphorylation of MAPK and NF-kB. The manuscript was well prepared. Some issues have to be revised before publishing.
1. In fig.5 A,B, the results of immunoblotting should be quantified.
2. In fig.5C, the inhibitory effect of JNK can also be included to support the claim that the main inhibitory effect of pseudane-VII was ERK pathway.
Author Response
RESPONSE TO REVIEWERS
Reviewer #2
This manuscript describes the anti-neuroinflammatory effect of pseudane-VII isolated from Pseudoalteromonas sp. M2 on LPS-induced microglia in vitro and in vivo. Pseudane-VII inhibited LPS-induced phosphorylation of MAPK and NF-kB. The manuscript was well prepared. Some issues have to be revised before publishing.
Comment 1: In fig.5 A,B, the results of immunoblotting should be quantified.
Response: We thank the reviewer for this helpful comment. We performed quantification of immunoblotting results and added it in Figure 5A and B.
Comment 2: In fig.5C, the inhibitory effect of JNK can also be included to support the claim that the main inhibitory effect of pseudane-VII was ERK pathway.
Response: We thank the reviewer for this helpful comment. We performed the inhibitory effect of JNK (inhibitor, SP600125) experiment as suggested by the reviewer and added results in Figure 5C.
Reviewer 3 Report
The manuscript by Kim et al describes psudane-VII from Pseudoalteromonas sp. suppresses neuroinflammation in lipopolysaccaride (LPS)-stimulated BV-2 microglial cells via attenuation of LPS-induced phosphorylation of MAPK and NF-κB. Administration of pseudane-VII in mice significantly reduced LPS-induced iNOS expression and microglia activation in brain. Pseudane-VII may be a potential novel target for treating neurodegenerative diseases.
The manuscript is well written and the results support the conclusions.
Comments:
1. The manuscript needs to be edited by native English speaker.
2. p-value has to be shown in Fig. 1B
Author Response
RESPONSE TO REVIEWERS
Reviewer #3
The manuscript by Kim et al describes psudane-VII from Pseudoalteromonas sp. suppresses neuroinflammation in lipopolysaccaride (LPS)-stimulated BV-2 microglial cells via attenuation of LPS-induced phosphorylation of MAPK and NF-κB. Administration of pseudane-VII in mice significantly reduced LPS-induced iNOS expression and microglia activation in brain. Pseudane-VII may be a potential novel target for treating neurodegenerative diseases.
The manuscript is well written and the results support the conclusions.
Comment 1: The manuscript needs to be edited by native English speaker.
Response: We thank the reviewer for this helpful comment. Our manuscript edited by www.biosciencewriters.com as native English speaker
Comment 2: p-value has to be shown in Fig. 1B
Response: We thank the reviewer for this helpful comment. We found that no cytotoxicity was shown and therefore no statistical significance was shown.
Round 2
Reviewer 1 Report
Dear authors, thank you for answering my comments.